# Unraveling Complex Hysteresis Phenomenon in 1,2-Dipalmitoyl-sn-Glycero-3-Phosphocholine Monolayer: Insight into Factors Influencing Surface Dynamics

**DOI:** 10.3390/ijms242216252

**Published:** 2023-11-13

**Authors:** Wisnu Arfian A. Sudjarwo, José L. Toca-Herrera

**Affiliations:** Institute of Biophysics, Department of Bionanosciences, University of Natural Resources and Life Sciences Vienna (BOKU), 1190 Vienna, Austria

**Keywords:** dissipated energy, DPPC monolayer, hysteresis, molecular packing, phase transition

## Abstract

This study explores the hysteresis phenomenon in DPPC (1,2-dipalmitoyl-sn-glycero-3-phosphocholine) monolayers, considering several variables, including temperature, compression and expansion rates, residence time, and subphase content. The investigation focuses on analyzing the influence of these variables on key indicators such as the π-A isotherm curve, loop area, and compression modulus. By employing the Langmuir–Blodgett technique, the findings reveal that all the examined factors significantly affect the aforementioned parameters. Notably, the hysteresis loop, representing dissipated energy, provides valuable insights into the monolayer’s viscoelasticity, molecular packing, phase transition changes, and resistance during the isocycle process. These findings contribute to a comprehensive understanding of the structural and dynamic properties of DPPC monolayers, offering insights into their behavior under varying conditions. Moreover, the knowledge gained from this study can aid in the development of precise models and strategies for controlling and manipulating monolayer properties, with potential applications in drug delivery systems, surface coatings, as well as further investigation into air penetration into alveoli and the blinking mechanism.

## 1. Introduction

The investigation of lipid monolayers has captivated scientists for decades due to their fundamental role in biological processes and their relevance in various technological applications [1]. One intriguing aspect of lipid monolayers is the phenomenon known as hysteresis. Hysteresis refers to the non-equilibrium behavior observed during the compression and expansion cycles of lipid monolayers. Specifically, it describes the differences in the surface pressure–area isotherms obtained during compression and expansion, indicating irreversible changes in the monolayer’s structure and properties [2,3,4,5]. This also represents energy dissipation, or energy lost within the monolayer during the cyclic process. The dissipative process involved in hysteresis can be attributed to molecular rearrangement or packing (e.g., phase transitions) [6,7].

The hysteresis of lipids is often observed in the study of pulmonary surfactants or in tear film lipids. In the case of the pulmonary system, the surfactants cover the alveolar surfaces of mammalian lungs, which are primarily composed of dipalmitoylphosphatidylcholine (DPPC) at around 31%. They play vital roles, including lowering surface tension to prevent atelectasis when breathing, eliminating pathogens, preventing the spread of pathogens, and controlling immune responses [8,9,10,11]. The lipid layer of the tear film is located on the front surface of the eye and serves as a protective barrier against excessive evaporation and foreign particles, while also contributing to the stability of the tear film. This lipid layer is structured with polar lipids, such as phospholipids, ceramides, and free fatty acids, which also function as surfactants. The tear lipids are constantly subjected to stress due to shear forces resulting from eye movements and the compression and expansion that occur during blinking. The compression and expansion of monolayers offer a valuable approach to simulating the impact of blinking on a specific tear film [6]. Consequently, understanding hysteresis can provide critical insights into the dynamics of breathing and respiratory mechanics on the lung surfactant monolayer or the blinking mechanism on the eyelids.

Furthermore, the study of lipid monolayers enables opportunities to create precise models and techniques for controlling monolayer properties. These advancements have potential applications in drug delivery systems, including the critical aspect of release kinetics. In this context, when a lipid monolayer is compressed, it can effectively function as a barrier, thereby slowing down the release of drugs. Conversely, expanding the monolayer facilitates faster drug release. This modulation of release kinetics can be finely tuned by adjusting the degree of compression or expansion, allowing for the achievement of specific release profiles over time. An example of a nanocarrier that illustrates this concept is the solid lipid nanoparticle, which consists of a lipid core enveloped by a surfactant monolayer. The controlled manipulation of lipid monolayers is integral to the design of such nanocarriers, ensuring that they release drugs in a manner that meets specific therapeutic requirements [12,13].

Experimental systems that aim to mimic the above mechanisms often involve the formation of Langmuir monolayers using amphiphilic compounds. Langmuir monolayers composed of phospholipids at the air–water interface are widely used as models for biological membranes, including tear lipid films and pulmonary surfactants. Among these phospholipids, DPPC is a well-known amphiphilic molecule that is commonly found in biological systems. These monolayers consist of a single layer of phospholipids or other amphiphiles, with their polar heads in contact with water and their hydrocarbon chains exposed to the air. When the monolayer is compressed or expanded while keeping the area constant or varying the temperature, it exhibits a series of hysteresis effects, causing changes in specific areas and exerting surface pressure [14,15].

Several studies have reported on the altered surface properties and phase behavior of lipids using the Langmuir–Blodgett technique due to variations in temperature [15,16,17], different chain lengths of acyl lipids [18,19,20], compositions with other lipids [21,22,23], mixtures with other components [24,25,26] or particles [27,28,29], differences in subphases [30,31,32,33], oxidation and degradation [34,35,36], and pH changes [37,38,39]. Only a few research cases have thoroughly investigated hysteresis using the same technique, particularly for phospholipids, e.g., [6,29,40,41,42,43,44]. Many articles discussing hysteresis studies are predominantly found in the context of mimicking meibomian lipid films [7,45,46,47]. A comprehensive exploration of hysteresis using the Langmuir–Blodgett technique has yet to fully emerge, to the best of our knowledge at the time of writing, thus leaving ample room for further investigation. Based on several parameters explored in DPPC studies, we will utilize these variables to investigate their relationship with the hysteresis effects of DPPC.

This manuscript aims to comprehensively investigate the hysteresis behavior of DPPC monolayers by focusing on the influence of temperature, compression–expansion cycling rate, and residence time. Temperature plays a crucial role in lipid monolayers’ behavior, affecting their fluidity, phase transitions, and molecular organization. Understanding the impact of temperature on the hysteresis response of DPPC monolayers is essential for unraveling their stability and functional properties. Additionally, the compression–expansion cycling rate simulates the mechanical forces experienced by monolayers during natural processes like blinking. By varying the cycling rate, one can analyze how different mechanical perturbation rates affect hysteresis behavior, providing insights into the dynamic behavior and stability of DPPC monolayers. Moreover, examining the effects of different residence times on the hysteresis response of DPPC monolayers allows us to understand the timescale at which memory effects occur and the extent to which properties are influenced by the duration of mechanical perturbation.

## 2. Results

Several factors affecting the hysteresis of DPPC monolayers, namely, temperature, low and high expansion rates, similar compression and expansion rates, residence time, and subphase, were investigated. Every factor was measured with three replications. The standard deviation was calculated from the replicated experiments.

### 2.1. Temperature

The first parameter measured was temperature to figure out the most appropriate condition for the other selected parameters. We studied five different temperatures, namely, 15 °C, 20 °C, 25 °C, 30 °C, and 35 °C, with a temperature deviation of ±1 °C. Figure 1 reveals the effect of temperature on the hysteresis area.

As observed, rising temperatures induce changes in the phase transition of DPPC. Especially, the plateau moves upward when temperature increases. Furthermore, the area of the plateau region (liquid-condensed–liquid-expanded; LC-LE) becomes smaller after ascending. Since the region of LC-LE decreases because of the rising temperature, this indeed modifies the hysteresis loop, even though the area of hysteresis is similar at every different temperature. Additionally, variations in conditions only minimally alter the value of the area per molecule. Regarding the molecular area of DPPC, the change in temperature has a slight impact. The areas of lipid molecules are 0.62 nm^2^, 0.62 nm^2^, 0.61 nm^2^, 0.62 nm^2^, and 0.69 nm^2^ for compression at 15 °C, 20 °C, 25 °C, 30 °C, and 35 °C, respectively. It is evident that all conditions have a nearly identical molecular area, except at 35 °C.

According to the presented figure, the plateau of the DPPC isotherm curve can be clearly seen for the experiment at 20–30 °C. For 15 °C, a plateau does not appear, while at 35 °C, the plateau almost diminishes. Due to this result, consideration is given to measuring every measurement at 20 ± 1 °C since it encompasses all regions of the DPPC phase transition. Furthermore, the region of coexistence (the plateau) at 20 °C is the largest and most visible of all the measurement conditions.

### 2.2. Low and High Expansion Rates

The hysteresis graph obtained from the π-A isotherm of the DPPC monolayer displays interesting characteristics, as illustrated in Figure 2. Despite being studied at various temperatures, the π-A isotherm curve of the DPPC monolayer generally agrees with the findings reported by reference [48]. To prevent collapse, the area per molecule (nm^2^/molecule) was intentionally maintained within the range of 1.000 to 0.485 nm^2^, ensuring a balance between compression and expansion. The maximum surface pressure achieved by all curves was approximately 54–55 mN/m. Interestingly, the liquid-condensed region remains unchanged regardless of variations in the rates of compression and expansion.

The hysteresis phenomenon is observed across all curves, as evident in the discrepancy between the compression and expansion curves. Furthermore, the plateau region of the compression curve differs from that of the expansion curve, gradually shifting downward, referring to the decreasing monolayer’s elasticity or ability to recover its original state after undergoing compression. For example, at compression and expansion rates of 25 mm/min and 5 mm/min (25/5 mm/min; Figure 2a), the liquid-condensed–liquid-expanded (LC-LE) phase shifts from a surface pressure of 5.0 mN/m to 4.4 mN/m at 0.75 nm^2^ area per molecule.

However, while the LC-LE moves downward due to compression–expansion forces, since the experiment is conducted continuously, one can distinguish that the rising rate of compression behaves in the opposite manner. With different expansion rates, increasing the compression rate causes the LC-LE phase to move upward. To illustrate, at a molecular area of 0.75 nm^2^, the point where the plateau region settles at 4.3 mN/m for a compression rate of 10 mm/min moves to 4.5 mN/m, 4.9 mN/m, and 5 mN/m for rates of 15 mm/min, 20 mm/min, and 25 mm/min, respectively. This behavior is investigated at an expansion rate of 5 mm/min. The resting expansion rate (25 mm/min) at the same point of surface pressure forces the LC-LE region to rise from 5.4 mN/m for a compression rate of 10 mm/min to 5.5 mN/m, 6.4 mN/m, and 7.2 mN/m for rates of 15 mm/min, 20 mm/min, and 25 mm/min, respectively.

However, surprisingly, even though the compression rate affects the plateau, it has a very small impact on the molecular area. For example, for rates of compression and expansion of 5/5, 10/5, 15/5, 20/5, and 25/5, the molecular areas of lipid are 0.58 nm^2^, 0.59 nm^2^, 0.59 nm^2^, 0.61 nm^2^, and 0.60 nm^2^ It seems that slower compression packs lipid molecules in a more orderly manner and tighter than faster compression. Therefore, the distance between both molecules’ nuclei is smaller. This also indicates that increasing the barrier approach speed has the consequence of moving away from equilibrium.

If one scrutinizes the inset, since the experiment was conducted continuously, the difference between both expansion rates becomes clear. The low rate allows lipid molecules to have more time for proper arrangement. Hence, every single curve appears distinct, and one can clearly see each curve separately. However, when we observe the other condition (high expansion rate), some curves appear stacked even though they can be distinguished from one another based on color differences.

A detailed analysis of hysteresis is shown in Figure 2c. Upon examining the bar graph, it becomes evident that the hysteresis area can be interpreted as dissipated energy of the cycle compression and expansion of the monolayer. In general, all areas of hysteresis at low expansion rates are larger than those at high expansion rates. For instance, at a compression rate of 5 mm/min, the hysteresis area at a low expansion rate is 6.58 × 10^−22^ J, whereas at a high expansion rate, it is 5.11 × 10^−22^ J. These values are smaller than thermal energy kT at 293 K (4.05 × 10^−21^ J).

Furthermore, a clear trend is observed where the hysteresis energy drops as the rate of compression increase. To demonstrate, when the expansion rate is 25 mm/min, the loop area of 5.11 × 10^−22^ J, attained at a compression rate of 5 mm/min, shrinks to 4.26 × 10^−22^ J, 2.75 × 10^−22^ J, 2.32 × 10^−22^ J, and 1.94 × 10^−22^ J for 10 mm/min, 15 mm/min, 20 mm/min, and 25 mm/min, respectively. The complete results can be found in the Appendix A (please see Appendix A). Based on these findings, we can find the linearity of the parameter for each expansion rate and define the linear equations as y = −5.49·10^−24^x + 6.85·10^−22^ and y = −1.59·10^−23^x + 5.85·10^−22^ for low and high expansion rates, respectively (equation y = ax + b; y: hysteresis energy, x: in this case relates to the compression rate and not a multiplication sign, b: energy intercept). The equation slope tells us the speed of the process of decreasing hysteresis energy during compression–expansion.

Another interesting aspect that can be extracted from different rates of expansion is the compression modulus, which is illustrated in Figure 3. The data selected for this analysis are solely from the compression force. The results align with the findings from [17,49], and have already been discussed. During compression, DPPC exhibits different Cs^−1^ ranging from 0 to 300 mN/m. The critical value of Cs^−1^ for the G-LE (gas-liquid-expanded) phase is below 12.5 mN/m, for the LE (liquid-expanded) phase it is in the range of 12.5 to 50 mN/m, for LS (liquid state) it is 50 to 100 mN/m, for the LC (condensed liquid) phase it is 100 to 250 mN/m, and for S (solid state) it is above 250 mN/m [49,50].The sharp minima appear as indicators of phase changing, from the LE phase to the coexistence phase, which happens at a surface pressure between 3 and 5 mN/m. The sharp minima of Cs^−1^ also represent the plateau region associated with the π-A curve.

Based on Figure 3a, at a low expansion rate (5 mm/min), we observe that an increasing compression rate affects the decrease in the compression modulus at surface pressures between 8 and 50 mN/m, in general. This implies that the lipids lose some of their characteristics in the liquid-condensed and solid phases, even though the change is negligible. This is because faster compression results in less tightly packed and dense molecules. The arrangement of molecules is more orderly when a slower compression speed is used. (An analogy is to imagine balls in a two-dimensional space that are compressed, reducing the space between them, with the possibility that they may not be evenly distributed. As a result, the solid properties of the packing are lower than if the balls were compressed at a slower rate, allowing each ball to achieve a more uniform and optimal distribution of positions) However, if we examine the low surface pressure range between 0 and 8 mN/m in more detail, each compression modulus for each rate appears stacked (big image). Nevertheless, the modulus values fluctuate significantly for low compression speeds and begin to decrease with increasing compression speed (see inset).

For the high-speed expansion case, in the big image in Figure 3b, the trend slightly changes. In the surface pressure range between 8 and 50 mN/m, the compression modulus curves have the appearance of being stacked on top of each other but are separated at low surface pressures between 0–8 mN/m. The sharp peak minima also shift to the right with increasing compression speed. This indicates that the change in the LE phase into the LE-LC phase slightly shifts. While there is a slight change in LE into the coexistence phase transition, the fluctuation in modulus values within this surface pressure range is lower for a low compression rate compared to that shown in Figure 3a.

One may see the compression modulus falling after reaching a surface pressure of 40 mN/m. This behavior indicates an abrupt change in the organization of lipid molecules, leading to them becoming more tightly packed. It signifies the transition from the liquid-condensed (LC) phase to the solid phase. Several studies have shown this phenomenon, which typically occurs at a surface pressure higher than 40 mN/m [51,52,53].

### 2.3. Similar Compression and Expansion Rates

After inspecting the impact of low and high expansion rates, it is worth considering the effects of similar compression and expansion rates. The result of this parameter is depicted in Figure 4. This study can identify the optimal rates that yield the desired properties and behaviors of the monolayer. In a nutshell, the appearance of the isocycle curve of the DPPC monolayer is similar to the results from the previous experiment. However, due to the unchanging DPPC monolayer, we can see a difference in the plateau region between this parameter and the previous ones. If one compares this figure to the previous figure (Figure 2), one conclusion can be drawn. When the expansion rate remains constant (Figure 2), increasing compression rates cause the plateau to move upward. Surprisingly, when compression and expansion rates are maintained at a similar level (Figure 4a), this plateau remains stable. Additionally, the shrinking area of the hysteresis loop can be recognized properly when the rate is lowered (see inset).

An inspection of the loop area was also conducted. The decrease in rates impacts the reduction in hysteresis. To be more specific, the areas of the loop are 6.58 × 10^−22^ J, 6.05 × 10^−22^ J, 4.55 × 10^−22^ J, 3.82 × 10^−22^ J, and 2.91 × 10^−22^ J for compression/expansion rates of 5/5 mm/min, 10 mm/min, 15 mm/min, 20 mm/min, and 25 mm/min, respectively. A decrease in area is a result of faster compression and expansion rates.

Figure 5 presents the compression modulus of DPPC at given rates. Every curve for different rates seems to be stacked along the selected surface pressure. The variation in modulus values at low surface pressure (0–8 mN/m) is lessened. One can still see that the increasing rate of compression and expansion narrows and reduces these fluctuations. In addition, there is no specific change in the compression modulus compared to the previous results shown in Figure 3. All conditions have similar results, including the sharp minima at a surface pressure of 3–5 mN/m, the LE-LC phase at a compression modulus of 50–100 mN/m, and liquid-condensed and solid phases at 100–250 mN/m and >250 mN/m, respectively. One appealing aspect of Figure 5 is the shape of its compression modulus curve. When one compares the two compression modulus curve shapes in the previous cases (high and low expansion rates), the curve in Figure 5 is a combination of both high and low expansion rates. At a low surface pressure, the compression modulus behaves similarly to the low expansion rate, but at higher surface pressure, its linearity resembles that of the high expansion rate.

### 2.4. Residence Time—Subphases

In this section, we aim to observe the impact of residence time or compression history on three different subphases, namely, PBS, water, and glucose, at a specific range of molecular area from 1 to 0.485 nm^2^. The observed residence time differences considered were 0, 1, and 2 min for each subphase, as illustrated in Figure 6. The complete illustration is provided in the Appendix A).

Regarding the subphase of the DPPC monolayer, we can evaluate its impact on the molecular area. The molecular area can be determined from the liquid-condensed phase (between 20 and 40 mN/m). By fitting a straight line from the LC phase, it is possible to obtain the area per molecule. In the case of DPPC spread on water, the molecular area is 0.59 nm^2^. This finding agrees with references [54,55]. However, it differs slightly when the subphase is changed to PBS or glucose. The molecular area of DPPC for both then becomes 0.61 nm^2^. This observation is consistent with several findings in references [33,56,57,58]. Consequently, changing the subphase from water to PBS or glucose leads to an increase in the molecular area.

Regarding the hysteresis area, no immediate changes in surface pressure are observed during decompression within the first 0 min after reaching the target surface pressure. However, with a residence time of 2 min, the surface pressure undergoes reciprocal sudden changes, resulting in a wider hysteresis loop that extends from the maximum pressure to the plateau region. Upon initial examination, the hysteresis area is broader for a 2 min residence time compared to a 0 min residence time. Across different subphases, no differences are observed in the region from the liquid-expanded (LE) to the liquid-condensed (LC) phases. However, the LC region is affected within a specific range of molecular area. For example, at an area per molecule of 0.485 nm^2^, both PBS and water yield surface pressures above 50 mN/m, while it is only around 40 mN/m for glucose. The addition of glucose reduces the interaction between lipid molecules. A higher surface pressure typically indicates a more stable monolayer configuration.

Upon examining the bar graph (Figure 7), it is evident that the energy dissipation increases for all subphases as the residence time lengthens from 0 min to 2 min. The hysteresis energy increases observed for PBS, water, and glucose are 50%, 104%, and 175%, respectively, for a 0 min to 2 min residence time. As a result, the dissipated energies of the DPPC monolayer on PBS, water, and glucose become 150%, 204%, and 175%, respectively, in comparison to the initial state without any residence time.

## 3. Discussion

In this current investigation, the results revealed what should be taken into consideration when dealing with DPPC, particularly hysteresis. Hysteresis relates to the molecular packing and reorganization of lipid monolayers. Our findings (see Table 1) highlight that one should give carefully consideration to the temperature of measurement, compression and expansion rates, residence time, and subphase.

Changing temperature has a significant impact on phase transition, and this has been studied and documented in [59,60,61]. Additionally, the experimental design discussed by Frey and Lee [62] has provided insight into the existence of a triple point (by changing temperature); the triple point lies at a temperature between 15 °C and 20 °C. Below the triple point (15 °C), the monolayer undergoes a phase transition from the gas phase to the condensed phase without passing through the LE phase. Conversely, above the triple point (T > 20 °C), the presence of a plateau region indicates a coexistence region between the liquid-expanded (LE) and liquid-condensed (LC) phases, which signifies a first-order phase transition.

Hysteresis is observed along the phase transition line, and therefore, changes in the gas region or the decline in the liquid-condensed line have an impact on the extent of the hysteresis loop. The lipid molecules are tightly packed due to stronger van der Waals forces between the alkyl chain and dipole–dipole interactions between head groups, resulting in a more condensed monolayer at the LC phase. Hence, the region of the DPPC phase transition changes, as seen in the reduction in the plateau area. Furthermore, as the temperature rises, the DPPC molecules gain thermal energy, leading to increased fluidity. The acyl chains of the lipid undergo more rapid and extensive motion, resulting in a less densely packed monolayer and a decrease in the hysteresis loop.

Remarkably, one overlooked parameter, namely, the rate of compression and expansion, influences significant changes in the properties of the DPPC isotherm curve. Changing the compression rates can enable the evaluation of the extent of reversibility within the monolayer [63]. When the DPPC monolayer is compressed at a low rate, the DPPC molecules have more time to rearrange and pack densely. This leads to stronger intermolecular interactions and a more ordered monolayer structure. The denser packing restricts molecular motion and increases resistance to compression. As a result, the monolayer exhibits a larger hysteresis area due to the energy dissipation and irreversibility during compression and decompression. On the other hand, when the compression rate is increased, the monolayer experiences rapid compression, leaving less time for the molecules to reorganize. The compressed state is less ordered, and the molecules may not have sufficient time to establish strong intermolecular interactions. This results in a more reversible compression–decompression process, as the monolayer can quickly return to its initial state. Therefore, a higher compression rate is expected to lead to a smaller hysteresis area. This result differs from the findings from the study of Xu and Zuo, where changing the compression rate does not support intercycle hysteresis, which is measured using Constrained Drop Surfactometry (CDS) [64].

The rate effect can also be seen from compression modulus values. When the monolayer is compressed at a low rate (expansion rate of 5 mm/min), the values of Cs^−1^ at high surface pressure are higher than those of the higher expansion rate, which suggests the creation of a densely packed monolayer. The values of Cs^−1^ will then drop with the increase in compression rate at low expansion rate, indicating decreasing molecular packing.

In real-life scenarios, the reversibility of molecular reorganization after compression does not always reach an equilibrium state. This process of molecular rearrangement requires time to occur. This phenomenon typically takes place at low surface pressure without losing any molecules in the subphase. Conversely, when tackling high surface pressure, the behavior may be irreversible. Therefore, it is crucial to determine the residence time or compression history to comprehend molecular packing accurately. It is primarily associated with the relaxation time of molecules on a specific surface. This differs slightly from previous studies that focused on the relaxation time at the same surface pressure (isobaric) [65,66].

When residence time increases at a molecular area of 0.485 nm^2^, the decrease in surface pressure is attributed to molecular organization after compression. This phenomenon was studied by Patino et al. [67], who observed associations between liquid-condensed and solid phases at surface pressures above 30 mN/m using Brewster Angle Microscopy. There is also a possibility of molecular loss when some clusters of lipids dive into an aqueous phase [68]. 

Regarding the change in the molecular area of DPPC, it is more likely that the interactions between DPPC molecules are affected by the subphases. The enlargement of the molecular area of DPPC indicates a lower intermolecular interaction with the PBS subphase compared to water. The (attractive) interaction of two molecules decreases when the distance between them increases. It appears that the intermolecular interaction on PBS is dominated by electrostatic forces. However, the increase in area per molecule is small; it is about 0.02 nm^2^/molecule. Thus, the area per molecule is bigger than that of water. In For glucose, we might have a different interaction mechanism, and DPPC is expected to have a stronger interaction with glucose compared to water, likely facilitated by hydrogen bonding. When the DPPC monolayer is compressed, glucose molecules penetrate the condensed monolayer [33]. Due to its larger size, glucose could hinder the mobility of lipid molecules and lead to a stronger interaction.

## 4. Materials and Methods

### 4.1. Materials

DPPC (1,2-dipalmitoyl-sn-glycero-3-phosphocoline) was purchased from Avanti Polar Lipid (Alabaster, AL, USA). PBS tablet (Phosphate Buffered Saline), methanol, and chloroform were from Carl ROTH (Karlsruhe, Germany). Glucose was supplied by Sigma Aldrich (Saint-Louis, MO, USA). Lipid solutions were prepared by dissolving them in chloroform/methanol (4:1, *v*/*v*) to obtain 2 mM, while PBS and glucose were dissolved in milli-Q water that had previously been purified using Millipak Gold 0.22 µm with final resistivity of 18.2 MΩ.cm. The final concentration of PBS and glucose was 10 mM. The lipid solutions would then be stored at −20 °C until further use.

### 4.2. Isocycle Experiment Parameters

The investigation of area per molecule (nm) affecting surface pressure (π) was conducted using a fully computerized Langmuir–Blodgett technique, as explained elsewhere [40,68]. The experimental setup involved a Langmuir monolayer on a mini–Langmuir Trough (KSV NIMA series, Espoo, Finland) of 24,300 mm^2^. Prior to the experiment, the trough was filled with a subphase consisting of 135–140 mL of PBS 1X (10 mM), pH 7.4. We installed two barriers and a piece of Wilhelmy paper in the trough. The Wilhelmy paper, which was 10.30 mm wide, was made from Whatman filter paper No. 93. The temperature of the subphase was controlled using a thermostat (MGW Lauda Krüss, Königshofen, Germany) with water circulation. Afterwards, 20 µL of DPPC solution was drawn using a Hamilton syringe and evenly spread dropwise onto the subphase between the two barriers. The system was then left to stabilize for 10 min to allow for solvent evaporation. The investigation focused on the influence of various parameters on hysteresis at 20 °C, excluding temperature itself. The area per molecule was the primary target and ranged from 1 nm^2^ to 0.485 nm^2^. All observations were replicated at least three times. The experimental parameters were as follows:Temperature. The temperature parameters were set to 15 °C, 20 °C, 25 °C, 30 °C, and 35 °C, with a temperature deviation of ±1 °C. The experiment was performed with compression and expansion rates of 25/25 mm/min.Low and high expansion rates. We compressed the barriers at rates ranging from 5 to 25 mm/min, with an interval of 5 mm/min. The lateral constant rate of compression was used a default mode. A low expansion rate of 5 mm/min and a high expansion rate of 25 mm/min were selected.Similar compression and expansion rates. The compression and expansion rates varied simultaneously, with values of 5 mm/min, 10 mm/min, 15 mm/min, 20 mm/min, and 25 mm/min, respectively. The resulting curve was then recorded.Residence time and subphase. To observe the compression history, we set different residence times after compression, namely, 0, 60, and 120 s (residence time refers to time that the barriers stop between the end of compression and start of expansion). The compression and expansion rates used were 25 mm/min, according to the previous measurement result. Two different subphases, water and glucose, were used as substitutes.

### 4.3. Data Analysis

Hysteresis Energy. A film’s resistance to deformation is linked to the extent of hysteresis investigated during the cyclic process. This occurs due to conformation changes, tilting, or reorientation in response to compression or expansion. In addition, it is a common attribute found in the imperfect elastic response to deformation of various biological materials. These structural rearrangements can lead to hysteresis (influencing the interactions between DPPC) during the compression–expansion cycle. During the isocyclic process, if rearrangements occur within the lipid monolayer, the shape of the compression curve will differ from that of the expansion curve. To measure the hysteresis energy as a function of hysteresis area, we employed R-Studio. The energy of hysteresis was determined as follows:
(1)$$\begin{array}{c} \mathrm{Hysteresis}\;\mathrm{Energy}\;(J)\;=\\ \left|\int area\;under\;compression\;curve-\int area\;under\;expansion\;curve\right| \end{array}$$

Compression Modulus. The compression modulus (Cs^−1^) was used to obtain information on the molecular packing and ordering of the DPPC monolayer [69]. To calculate the compression modulus, the equation below was employed:
(2)$${Cs}^{-1}\;=\;-A_{0}(\frac{d\pi }{dA})$$
where A is area per molecule and π is the surface pressure respective to the area.

## 5. Conclusions

The experimental results of the observation of the DPPC monolayer bring deeper insight to the understanding of the hysteresis phenomenon. They reveal that each parameter has a significant impact on hysteresis behavior, as summarized in Table 1.

Specifically, the effect of temperature was primarily associated with the phase transition behavior, which was evident from the plateau in the isotherm curves. This step is crucial for deciding the proper condition of the rest measurement. The compression and expansion rates played a crucial role in determining the magnitude of hysteresis, reflecting the response of DPPC molecules to deformation during isocycle processes. Moreover, the residence time influenced the uniformity and orderliness of the molecular arrangement under compression, as well as the interaction between DPPC head groups and the subphase.

In conclusion, this study provides a comprehensive understanding of the hysteresis phenomenon in DPPC monolayers and highlights the influence of various parameters. The findings shed light on the dissipated energy and dynamic and compression moduli, and their dependence on temperature, compression and expansion rates, residence time, and subphase properties. This research contributes to the broader understanding of monolayer behavior and has implications for applications in surface science, drug delivery systems, and biomembrane research. Further investigations can explore the implications of these findings in relevant biological and technological systems.

## Figures and Tables

**Figure 1 ijms-24-16252-f001:**
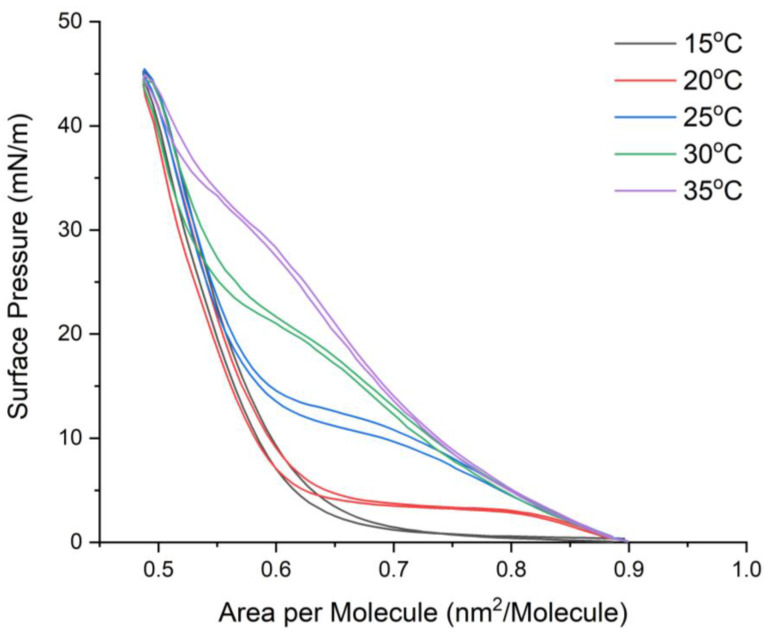
The π-A isocycle curve of the DPPC monolayer at different temperatures. Temperature affects the liquid-expanded–liquid-condensed coexistence phase (plateau). The plateau disappears when the temperature increases.

**Figure 2 ijms-24-16252-f002:**
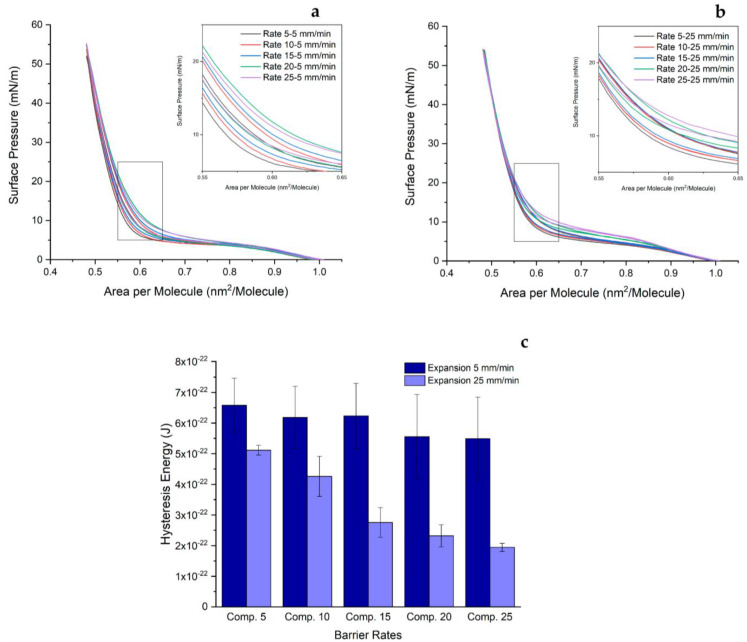
The π-A isotherm isocycle curve of the DPPC monolayer at (**a**) an expansion rate of 5 mm/min, (**b**) an expansion rate of 25 mm/min. The hysteresis area of every isocycle curve is presented in (**c**). Both insets are the captured isocycle curves at an area per molecule of about 0.55–0.65 nm^2^ and a surface pressure of 0–25 mN/m. These experiments were conducted at 20 ± 1 °C. The area delimited by the curves provides the energy lost due to hysteresis.

**Figure 3 ijms-24-16252-f003:**
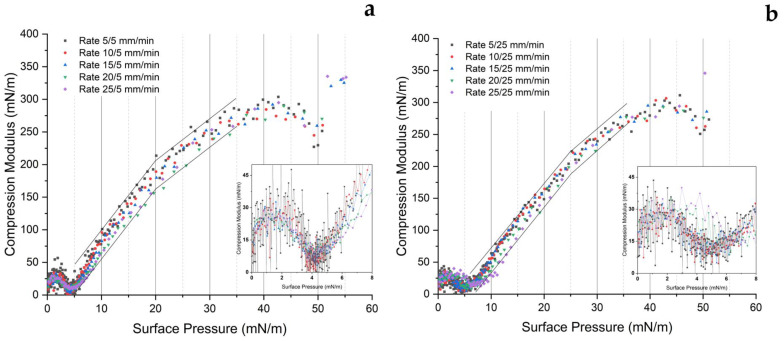
Compression modulus (Cs^−1^)–surface pressure (π) curves of DPPC monolayer: (**a**) at an expansion rate of 5 mm/min, and (**b**) at an expansion rate of 25 mm/min. Both insets are the captured area of the curve at a surface pressure of 0–8 mN/m and compression modulus of 0–50 mN/m, with a connected line on every point of the modulus. These experiments were carried out at 20 ± 1 °C.

**Figure 4 ijms-24-16252-f004:**
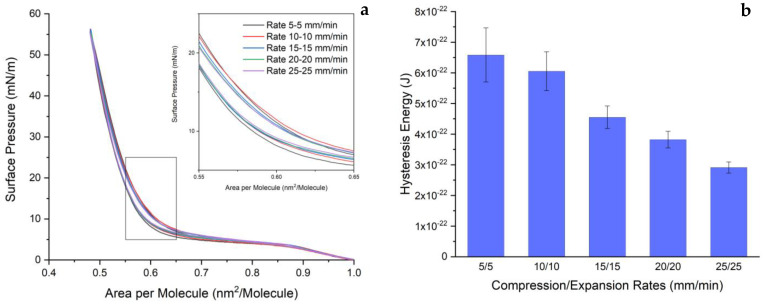
(**a**) The π-A isocycle curve of DPPC monolayer with similar compression and expansion rates ranging from 5/5 mm/min to 25/25 mm/min. (**b**) The hysteresis loop area obtained from the isocycle curve. The inset shows the captured isocycle curves at an area per molecule between 0.55 and 0.65 nm^2^, and surface pressure between 0 and 25 mN/m. These experiments were conducted at 20 ± 1 °C.

**Figure 5 ijms-24-16252-f005:**
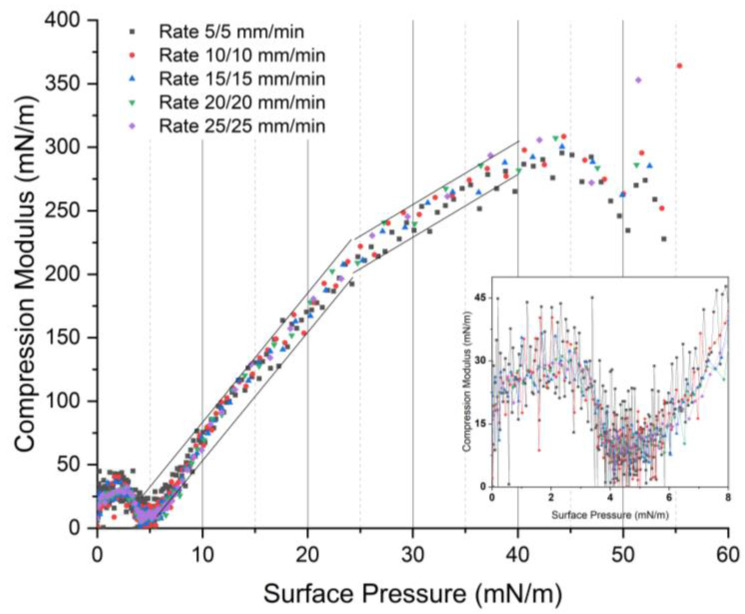
The compression modulus (Cs^−1^)-surface pressure (π) curves of DPPC monolayer for similar compression and expansion rates. These experiments were carried out at 20 ± 1 °C.

**Figure 6 ijms-24-16252-f006:**
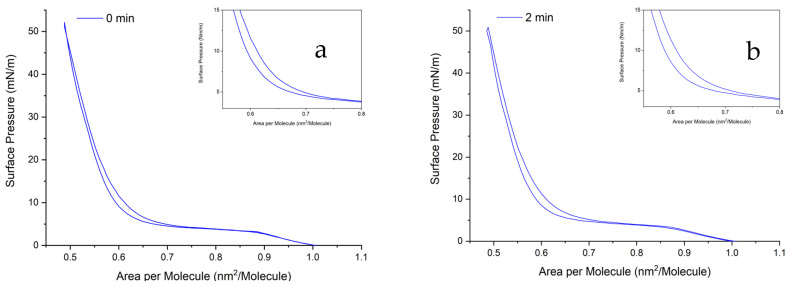
π-A isocycle curves of DPPC monolayer. (**a**,**b**) π-A isotherm curve of DPPC on PBS 1X, (**c**,**d**) π-A isotherm curve of DPPC on water, and (**e**,**f**) hysteresis area of the π-A isotherm curve of DPPC on glucose 10 mM. (**a**,**c**,**e**) Residence time of 0 min; (**b**,**d**,**f**) residence time of 2 min. These experiments were carried out at 20 ± 1 °C.

**Figure 7 ijms-24-16252-f007:**
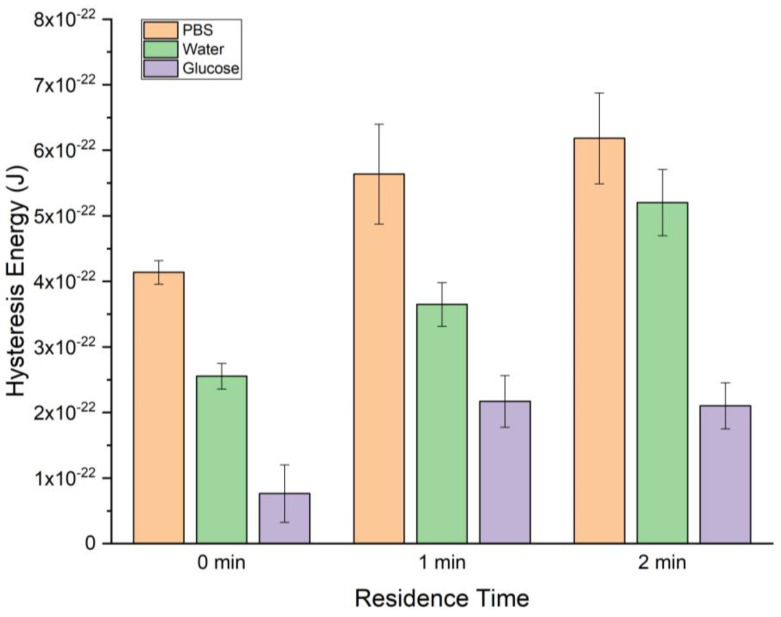
The hysteresis area of the π-A isotherm curve of DPPC for different residence times, namely, 0, 1, and 2 min. These experiments were conducted at 20 ± 1 °C.

**Table 1 ijms-24-16252-t001:** Summary of parameter changes in the isocycle of DPPC monolayers.

Parameters	Conclusion
Temperature	- The position of the plateau (LE-LC phase) changes. - The length of the plateau changes.
Low expansion rate	- The position of LC phase changes. - Hysteresis energy decreases with an increase in compression rate. - The compression modulus remains constant at low values. - There is a small impact on the area per molecule.
High expansion rate	- The position of the plateau (LE-LC phase) shifts. - Hysteresis energy decreases with an increase in compression rate. - The sharp peak minima shift to the right with an increase in the compression rate.
Similar compression and expansion rates	- The plateau position remains stable. - Hysteresis energy drops with increasing rates. - The compression modulus adopts the behavior of both low and high expansion rates, i.e., stable at low and high surface pressures.
Residence time	- Hysteresis energy increases when there is an increase in residence time. - The position of the plateau remains constant.
Subphase	- The area per molecule changes. - The position of the plateau is stable, but the hysteresis differs. - Based on similar area per molecule, the surface pressure shifts accordingly.

## Data Availability

The data presented in this study are available on request from the corresponding authors.

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
