# Peer review of "Unraveling Complex Hysteresis Phenomenon in 1,2-Dipalmitoyl-sn-Glycero-3-Phosphocholine Monolayer: Insight into Factors Influencing Surface Dynamics"

_ijms, 2023, doi:10.3390/ijms242216252_

Round 1
Reviewer 1 Report
Comments and Suggestions for Authors
The paper presents an extensive investigation of the several parameters on the hysteresis shown during compression and expansion of DPCC monolayers. The subject is consistently set into the framework of current research on the subject, the investigation methods satisfactorily described and the results discussed extensively, with conclusions summarised at the end. This reviewer feels therefore that the paper can be accepted in its current form. As minor remarks:
- it may be worth indicating the uncertainty in temperature during the tests;
- the triple point temperature should be clearly stated: from r.315-317 it can only be concluded it lies somewhere between 15 and 20 °C;
- the units for the y-axis caption in Fig. 1 are typed wrongly (Nm/m instead of mN/m).
Comments on the Quality of English LanguageSection 3 should be checked for language.
Author Response
We thank the reviewer for the constructive remarks.
Question: it may be worth indicating the uncertainty in temperature during the tests.
Answer: We thank the referee for his/her comment. We have indicated the uncertainty (1°C) of the temperature in the revised text (it is highlighted in yellow).
Question: the triple point temperature should be clearly stated: from r.315-317 it can only be concluded it lies somewhere between 15 and 20 °C;
Answer: Thank you for the comment. According to Frey and Lee (Langmuir 2007, 23, 2631–2637, doi:10.1021/la0626398) the triple point lies at temperature between 15°C and 20°C. This comment is added in the revised version.
Question: the units for the y-axis caption in Fig. 1 are typed wrongly (Nm/m instead of mN/m).
Answer: We thank the referee for the comment. The units have been corrected. A new figure has been added.
Reviewer 2 Report
Comments and Suggestions for Authors
Reviewer report on Manuscript Draft ‘Unraveling the Complex Hysteresis Phenomenon in DPPC Monolayer: Insight into Factors Influencing Surface Dynamics’.
In this work authors explores the hysteresis phenomenon in DPPC (1,2-dipalmitoyl-sn-glycero-3-phosphocholine) monolayers, considering several variables including temperature, compression and expansion rates, residence time, and subphase content. The knowledge gained from this study can aid in the development of precise models and strategies for controlling and manipulating monolayer properties, with potential applications in drug delivery systems, surface coatings, as well as further investigation into air penetration into alveoli and the blinking mechanism.
This investigation is interesting, from the point of view of physical chemistry, and material science. The research is in the scope of the journal. Therefore, the manuscript can be published after some minor corrections and improvements:
It is remarkable that the knowledge gained from this study can aid in the development of precise models and strategies for controlling and manipulating monolayer properties, with potential applications in drug delivery systems. Therefore, Introduction could be supported by justification of importance of drug delivery systems (Development of essential oil delivery systems by ‘click chemistry’ methods: possible ways to manage Duchenne muscular dystrophy. Materials, 2023, 16, 6537.).
Table 1 from conclusions could be moved into Discussion part.
The importance of the research for the modelling of drug delivery systems could be also reflected in Conclusion section.
Comments on the Quality of English Language
Minor editing of English language required.
Author Response
We thank the referee for his/her valuable comments. We have introduced the suggested changes in the revised manuscript.
Question: It is remarkable that the knowledge gained from this study can aid in the development of precise models and strategies for controlling and manipulating monolayer properties, with potential applications in drug delivery systems. Therefore, Introduction could be supported by justification of importance of drug delivery systems (Development of essential oil delivery systems by ‘click chemistry’ methods: possible ways to manage Duchenne muscular dystrophy. Materials, 2023, 16, 6537.).
Answer: We thank the referee for the comment. Following his/her suggestion we have written a new paragraph in the introduction (including the proposed reference) in the revised manuscript. The text is highlighted in yellow.
Question: Table 1 from conclusions could be moved into Discussion part.
Answer: Thank you for the comment. We have moved Table 1 to the discussion section as suggested by the referee.
Question: The importance of the research for the modelling of drug delivery systems could be also reflected in Conclusion section.
Answer: Thank you for the comment. We have addressed the suggested comment in the conclusion part. The text is highlighted in yellow.